# Real-World Data about Commonly Used Antibiotics in Long-Term Care Homes in Australia from 2016 to 2019

**DOI:** 10.3390/antibiotics12091393

**Published:** 2023-08-31

**Authors:** Chloé C. H. Smit, Kris Rogers, Hamish Robertson, Katja Taxis, Lisa G. Pont

**Affiliations:** 1Graduate School of Health, University of Technology Sydney, Sydney, NSW 2008, Australia; 2School of Public Health & Social Work, Queensland University of Technology, Brisbane, QLD 4000, Australia; 3Department of PharmacoTherapy, Epidemiology and Economics, Faculty of Science and Engineering, University of Groningen, 9713 AV Groningen, The Netherlands

**Keywords:** antibiotics, urinary tract infections, long-term care, drug utilization, duration of therapy

## Abstract

In this study, we use real-world data to explore trends in antibiotic use in a dynamic cohort of long-term care (LTC) residents. A cross-sectional retrospective analysis of pharmacy medication supply records of 3459 LTC residents was conducted from 31 May 2016 to 31 May 2019. The primary outcome was the monthly prevalence of residents with an antibiotic episode. Secondary outcomes were the type of antibiotic used and duration of use. Over the three-year study period, residents were supplied 10460 antibiotics. On average, 18.9% of residents received an antibiotic monthly. Antibiotic use decreased slightly over time with a mean of 168/1000 (95% CI 146–177) residents using at least one antibiotic per month in June 2016 to 148/1000 (95% CI 127–156) in May 2019. The total number of antibiotic days per 100 resident days remained relatively constant over the study period: 8.8 days in 2016–2017, 8.4 in 2017–2018 and 6.4 in 2018–2019. Prolonged durations exceeding 100 days were seen for a small percentage of residents. We found extensive antibiotic use, which is a recognized contributor to antimicrobial resistance development, underscoring the necessity for quality treatment guidelines in this vulnerable population.

## 1. Introduction

The development and spread of antimicrobial resistance (AMR) are a global issue. In the long-term care (LTC) setting, previous research has found that up to 35% of LTC residents are colonized with one or more multi-resistant bacteria [1]. Inappropriate antibiotic use, including overuse, use for inappropriate indications, poor antibiotic choice, and suboptimal antibiotic dose and/or duration of use, is considered a key modifiable risk factor for AMR development [2].

Overall antibiotic use in the LTC setting is high, with 62% of residents receiving one or more antibiotics annually [3,4]. Older persons living in LTC homes are frail and have multiple comorbidities, cognitive impairment and compromised immune systems, all of which can increase the risk of infection [5]. Infection rates among LTC residents are high varying from 1.5 to 9.5 infections per 1000 occupied bed days [6]. Urinary tract infections (UTI) and respiratory tract infections (RTI) are the most common infections among LTC residents [5]. These infections are commonly managed with antibiotics either as treatment or prophylaxis and account for up to 60% of all antibiotics prescribed in LTC homes [3,7,8,9,10,11,12].

As antimicrobial resistance develops over time, longitudinal drug utilization research exploring antibiotic use in the LTC setting will aid health professionals and policymakers in identifying potential areas of suboptimal use that may drive the development of antimicrobial resistance. To date, much of the drug utilization research around antibiotics in the LTC setting has focused on point prevalence, rather than consideration of longitudinal trends [7,9,13,14]. Of the few longitudinal studies that have been conducted in the LTC setting either the population included was small [6], or the time period was limited to a one-year follow-up, reducing their value in the identification of trends that may contribute to the development of antimicrobial resistance over time [15,16]. Two studies exploring trends in antibiotic use in LTC over a longer time using real-world data have been conducted. However, these studies reported conflicting results, with an Australian study finding a 5% decrease in antibiotic use by LTC residents over a four-year period [17], while a larger Canadian study found no change in antibiotic use over the same time period [18]. To our knowledge, no studies exploring the duration of antibiotic use in LTC were identified despite this being a key consideration in the development of AMR.

With concerns around high and potentially inappropriate antibiotic use driving AMR, understanding how antibiotics are used in the LTC setting is important for health professionals and policymakers alike. Real-world data provides the opportunity to understand contemporary “real world” practices around antibiotic use. Therefore, the aim of this study is to use real-world data to investigate trends, types of antibiotics used and duration of antibiotic use among LTC residents.

## 2. Results

### 2.1. Cohort Characteristics

A total of 3416 unique residents were present in the cohort for a total of 1,767,788 resident days between 31 May 2016 and 31 May 2019. Resident characteristics are shown in Table 1.

Over the three-year period, residents used a total of 10460 antibiotic episodes of 31 different types of antibiotics (Table 2). Antibiotics least frequently used were ceftriaxone (6 episodes); fusidic acid (6 episodes); tinidazole (5 episodes); minocycline (4 episodes); cefazolin (3 episodes); moxifloxacin (3 episodes); azithromycin (2 episodes); and procaine-benzylpenicillin (2 episodes). Ceftazidime, ertapenem and fosfomycin were all only used once.

### 2.2. Monthly Trends of All Systemic Antibiotic Episodes 

Our results showed seasonal trends in the use of all antibiotics, with antibiotics less frequently initiated in summer and increasing use in winter (Figure 1). Use of all antibiotics was highest in May 2017, with 229 residents per 1000 residents (95% CI 200–221) using any antibiotic and lowest around February 2018 (158 per 1000 residents, 95% CI 158–177). From September 2018 onwards, the use of all antibiotics stabilized to around 187 residents using an antibiotic per 1000 residents (95% CI 182–202) per month.

Across the total study period, on average 189 residents per 1000 residents received a systemic antibiotic each month.

### 2.3. Monthly Trends per Individual Antibiotic

Monthly trends of the ten most frequently used antibiotics are shown in Figure 2. Trends for other types of antibiotics were not analyzed due to an insufficient amount of data. Cefalexin was most frequently used throughout the study period, with the highest number of antibiotic episodes in May 2017 (86/1000 residents) and the lowest in February 2018 (61/1000 residents). Both amoxicillin with and without clavulanic acid were commonly used with an average of 21/1000 residents receiving these antibiotics per month. Amoxicillin had the highest use in August 2016 (34/1000 residents), July 2017 (28/1000 residents) and July 2018 (32/1000 residents), whereas amoxicillin with clavulanic acid was additionally high in January 2017 (31/1000 residents). Doxycycline did not show seasonal patterns and was most frequently used in July 2017 (45/1000 residents), August 2018 (35/1000 residents) and February 2019 (35/1000 residents). Flucloxacillin use was low with a slight increase in February 2017 (27/1000 residents). The monthly average of ciprofloxacin, clindamycin, roxithromycin and trimethoprim with sulfamethoxazole was low (all around 7/1000 residents). The agents most frequently used were moderate-spectrum antibiotics cefalexin and amoxicillin, followed by more broad-spectrum antibiotics doxycycline, amoxicillin with clavulanic acid and trimethoprim.

### 2.4. Duration of Use 

The total number of antibiotic days per 100 resident days remained relatively constant over the study period: 8.8 days in 2016–2017, 8.4 in 2017–2018 and 6.4 in 2018–2019. The majority of antibiotic episodes were used for 6 to 14 days (80.5%, *n* = 9709) followed by durations up to 5 days (8.7%, *n* = 856) and durations from 15 days up to 3 months (9.2%, *n* = 904). Only a small proportion of the antibiotic episodes was used for 3–12 months and over 12 months (1.3%, *n* = 12 and 0.3%, *n* = 30 respectively). Amongst these, antibiotics were used for an average of 261 days.

Looking at the duration of use per type of antibiotic, the median duration of use was 8 days for the 10 most frequently used antibiotics (Figure 3). There was insufficient data to obtain information on the duration of use for the other types of antibiotics. Sulfamethoxazole with trimethoprim had a longer median duration of 10 days whereas trimethoprim, roxithromycin and amoxicillin with clavulanic acid had a shorter duration of 7 days. Cefalexin, amoxicillin and trimethoprim had 2 distinct duration distributions around the 6–8 days. Prolonged durations exceeding 100 days were seen for most antibiotics, with the exception of clindamycin.

## 3. Discussion

This study demonstrated the value of real-world data in drug utilization research. Our data showed high antibiotic use among LTC residents with one in five residents using an antibiotic each month. Across the study period, the use of all antibiotics decreased slightly over time and overall utilization of moderate-spectrum and broad-spectrum antibiotics was higher than that of narrow-spectrum agents. While we found that the majority of antibiotics were used for durations of around 1 week, prolonged utilization exceeding three months did occur.

In this study, we found relatively high antibiotic use among LTC residents. This is probably due to the longitudinal design of this study, compared to point prevalence or incidence studies. The prevalence of antibiotic use among LTC residents appears country-specific and our prevalence of 18.8% of residents is considerably higher than that reported in European LTC homes. A European study looking at the monthly prevalence of residents receiving an antimicrobial agent across 19 European countries between April and November reported that approximately 6% of LTC residents used an antibiotic each month [19]. However, it should be noted that this study was conducted during the European summer months, and, given the seasonal variation found in our study, lower utilization over summer is expected. The period over which the prevalence is measured will also contribute to variation between estimates, with previous research reporting point prevalence estimates ranging from 3% to 11% and period prevalence varying from 44.9% to 77.8% [3]. Our findings are also considerably higher than those reported in other Australian studies with the annual Aged Care National Antimicrobial Prescribing Survey (AC NAPS) reporting single-day point prevalences of 5.8% in 2016, 5.3% in 2017 and 5.5% for both 2018 and 2019 [5]. The decline in antibiotic use from 2016 to 2017 and stabilization of use from 2018 to 2019 reported by the AC NAPS is similar to those observed in our research and may be a result of the establishment of the Antimicrobial Use and Resistance in Australia (AURA) surveillance project and other initiatives, such as those from NPS MedicineWise aiming to reduce unnecessary antibiotic utilization [5,20].

Frequent use of broad-spectrum antibiotics affects the bacterial flora and puts selective pressure on it, which increases the development of multi-resistant bacteria [21]. In our research moderate- to broad-spectrum antibiotics, namely cefalexin, amoxicillin and amoxicillin with clavulanic acid were the most used antibiotics among those studied, raising concerns regarding AMR. In Australia, most antibiotics are reimbursed by the government as part of the Pharmaceutical Benefits Scheme [22]. To support the appropriate use of antibiotics, strict prescribing criteria including a limited quantity of supply and limited indications for use exist for a number of antibiotics [23]. The low use of fluoroquinolones (ciprofloxacin and norfloxacin) observed in our study is likely to be directly related to strict limitations on reimbursement of these agents [22]. Of note is that antibiotics marked as WATCH antibiotics by the WHO were not frequently used, which may also be due to Australian policies and prescribing restrictions [24]. Ultimately, prevention of infections in this setting is important to prevent antibiotic use and resistance, with measurements combining technical and socio-adaptive techniques described by Mody et al. showing success in reducing catheter-associated UTI [25]. We observed a possible trend toward reduced number of antibiotic days per 100 resident days from 2016 to 2019. Future longitudinal research over a longer time frame is needed to explore these trends further. We found that most antibiotic episodes were of a short duration. This appears to be consistent with Australian guidelines which indicate short-term use for the majority of acute infections where an antibiotic is required [12]. Similar findings regarding the duration of use of antibiotics for management of UTI in LTC have been reported in two Canadian and one US studies where the majority of antibiotic prescribing was short term for less than 14 days [8,18,26]. These findings indicate the success of practice guidelines in guiding rational antibiotic prescribing practices.

In this study, while most episodes were of short duration, a small percentage of LTC residents were using antibiotics for extended durations of longer than 100 days. This was seen for all antibiotics except for clindamycin. Longer durations of use may indicate the use of antibiotics for prophylaxis or prevention of infection rather than treatment of an acute infection. Cefalexin was predominantly used for prolonged duration and the national Australian treatment guidelines do recommend low-dose cefalexin, nitrofurantoin or trimethoprim for prophylaxis for up to 6 months duration [12]. The prolonged duration of use in this study was considerably lower than that reported by Daneman et al., who found that 21% of LTC residents were on antibiotics for longer than 90 days [8]. This higher prevalence is most likely due to differences in the type of antibiotics included with Daneman et al including all systemic antibiotics and our research focused only on those frequently used.

### Strengths and Limitations

A key strength of this study was the use of real-world data to examine antibiotic utilization in a large population of Australian LTC residents. We were able to explore utilization patterns over a 3-year period and examine trends in the prevalence and duration of the antibiotics most frequently used. Using real-world data provides valuable information to health professionals and policymakers to enable therapeutic decision making and the development of robust health policy.

There were several limitations that should be considered when interpreting the findings from this study. While detailed information on the choice and duration of antibiotics supplied at the individual resident level was available, the dataset consisting of pharmacy records, does not contain any information on the indication or infection site, type or organism for which the antibiotic was prescribed. Additionally, there were no pathology results available on the sensitivity of bacterial isolates from urine, blood or fecal cultures before or after antibiotics were used. We therefore did not have data on the evolution, cure or failure of antibiotic use. Higher antibiotic use during the winter season suggests use for indications related to seasonal influences such as respiratory tract infections.

We found high antibiotic utilization among LTC residents and wide use of broad-spectrum agents, both of which may contribute to the development of AMR. Our findings highlighted the importance that high-quality treatment guidelines have in guiding prescribing and supporting the rational use of medicines. Finally, this research illustrates the value of using real-world data in drug utilization research, providing important information on current practice to understand the way antibiotics are used within fragile populations with the ability to support policy and practice in ensuring optimal antibiotic use.

## 4. Materials and Methods

### 4.1. Study Design

A retrospective, longitudinal repeated monthly cross-sectional analysis of antibiotic utilization among LTC residents over a 3-year period between 31 May 2016 and 31 May 2019 was conducted.

### 4.2. Setting and Population

The study was conducted in a dynamic cohort of LTC residents in the Illawarra region of Australia. All LTC residents who received one or more medications from a pharmacy contributing to the pharmacy medication supply dataset described below between 31st May 2016 and 31st May 2019 were included in the analysis.

### 4.3. Data Source

The dataset for this study comprised pharmacy Dose Administration Aids (DAA) medication records for medications supplied to residents in LTC. Residents of LTC facilities in Australia are predominantly 65 and over, with most residents in the 85–89-year-old group at admission [27]. In Australian LTC, all medications used by residents, even those that can be purchased without a prescription, must be prescribed by a general practitioner or other authorized health professional prescribers. Medicines prescribed in LTC are supplied by community pharmacies, with a single community pharmacy generally supplying medications for all residents within a single LTC home. To facilitate medication management within the LTC, medications for LTC residents are provided by the community pharmacy pre-packed for individual residents in weekly DAA [28]. Medications, such as liquids or inhalers, which are not physically packed in a DAA are still recorded in the DAA software, and the DAA dataset contains complete weekly medication supply records for included LTC residents. DAA medication records provide a responsive, real-world data source that has been used to explore a range of drug utilization questions [28,29,30,31,32].

Variables in the data source include pharmacy code, facility code, resident code, resident age, resident status indicating if residents are present in the facility, hospitalized, on holiday, moved or passed away, medication brand name, medication strength, medication directions, medication start date and medication cease date for all medications supplied to LTC residents by the pharmacy.

### 4.4. Definitions

#### 4.4.1. Antibiotics

Medications were coded using the WHO Anatomical Therapeutic Chemical (ATC) classification [33]. All systemic antibiotics (ATC level 2: J01) available in Australia during the study period were included in the analysis.

#### 4.4.2. Antibiotic Episodes 

An antibiotic episode was defined as the number of consecutive days between the date that an antibiotic was commenced and the date the antibiotic was ceased. As per Daneman et al, a new antibiotic episode was considered to have commenced if there was a gap of more than 3 days between a prior cease date and the next start for the same antibiotic [8].

### 4.5. Follow-Up Time

We calculated the time of follow-up for each resident stratified into calendar years, as the number of days between the first start date for any medication appearing in the dataset and the last cease date of any medication per resident (Appendix A). For residents who were flagged as having an active status with an end date of 31st May 2019, the end date of data extraction was used instead of the last cease date in calculating follow-up time.

To determine the number of calendar days that each resident contributed to the dataset, and to compare the calendar years, the number of resident days was determined as the sum of time of follow-up for all the residents per 1000 residents’ days.

Resident age was determined on the 1st of June of each calendar year.

### 4.6. Analysis

#### 4.6.1. Monthly Prevalence of Residents with One or More Antibiotic Episodes

The monthly prevalence of all antibiotic use was calculated as the number of residents with one or more antibiotic episodes for that month divided by the total number of residents present in the LTCs in that month and expressed per 1000 residents (Appendix A). The prevalence of use of antibiotics commonly used for the management of urinary tract infections was calculated in the same matter. Results were plotted in a scatterplot with 95% confidence intervals.

#### 4.6.2. Duration of Antibiotic Use

The duration of use was calculated for each individual antibiotic episode by subtracting the last date of each antibiotic episode from the first date of the episode (Appendix A). Duration of use was explored using three metrics: duration per three months (100 days), percentage of antibiotic users per antibiotic duration of use category and mean duration per antibiotic type.

To determine the average number of days of antibiotic therapy per 100 resident days per calendar year, the total duration of antibiotic use was summed and divided by the total follow-up time in days.

Duration of antibiotic episodes was stratified into 5 duration levels: ≤5 days; 6–14 days; 15–30 days; 31–90 days; 91–365 days and >365 days over the total study period. This was done to see how many of the antibiotic episodes were used for prolonged durations. The percentage of episodes in each duration level was calculated as the number of antibiotic episodes used in the duration level divided by the total number of antibiotic episodes in our dataset. 

The mean duration of use per antibiotic type over the three-year period was presented in Violin plots on a log scale.

All analyses were conducted using the R studio statistical package R 1.4.1106 (R Foundation for Statistical Computing, Vienna, Austria).

## Figures and Tables

**Figure 1 antibiotics-12-01393-f001:**
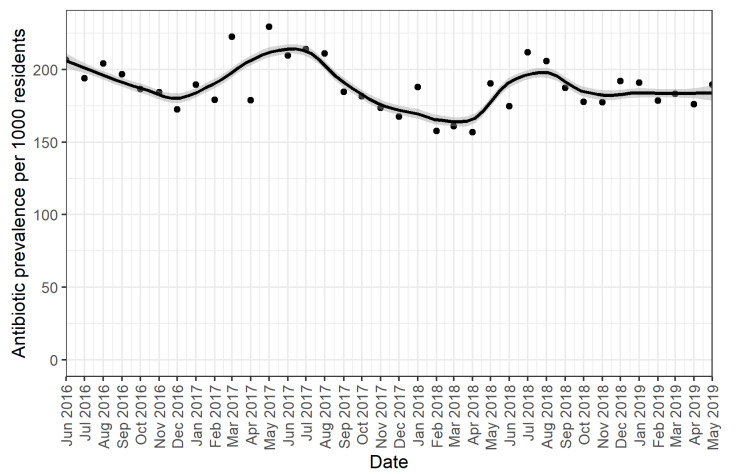
Monthly prevalence per 1000 residents of systemic antibiotics. The dots represent the monthly prevalence of antibiotic use per resident present in the LTC facilities per 1000 residents. The grey lines indicate 95% confidence intervals around the monthly estimates.

**Figure 2 antibiotics-12-01393-f002:**
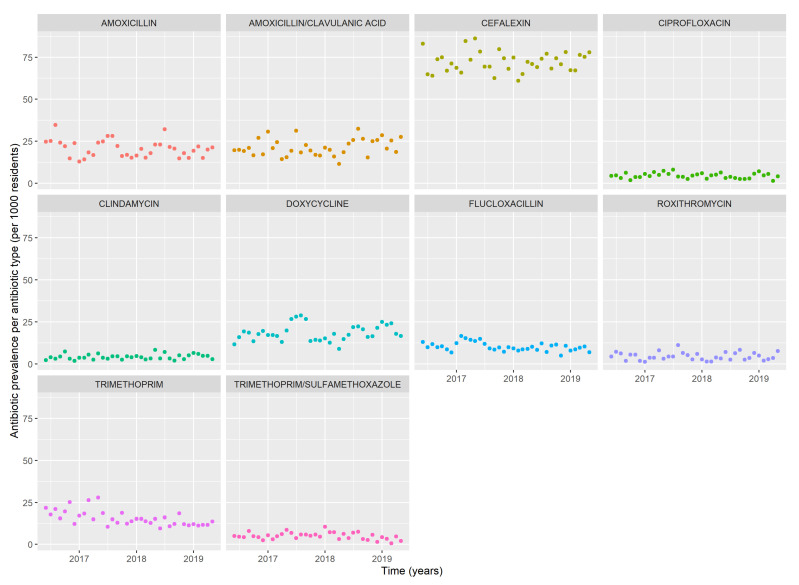
Monthly prevalence of antibiotic episodes per 1000 residents of the 10 most frequently used antibiotics.

**Figure 3 antibiotics-12-01393-f003:**
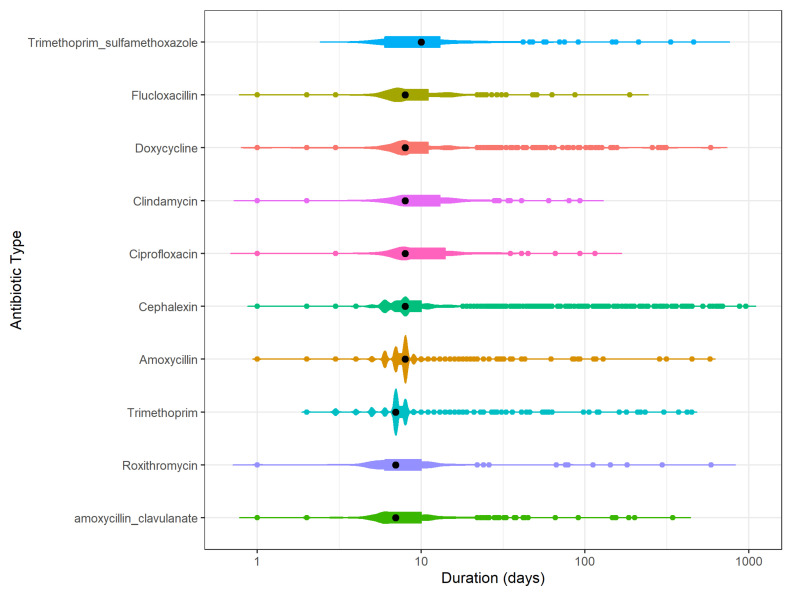
Number of antibiotic episodes with a particular duration of use per antibiotic type for the top 10 most frequently used antibiotics. From longer median duration (black dot) at the top to shorter median duration at the bottom. The colored dots indicate one antibiotic episode for that duration of use of one antibiotic type. Violin plot shows the distribution of the supplies. The longer the line of the violin plot the more the duration of use varies for that antibiotic type. The thicker the violin plot the more frequently an antibiotic is used for that duration. Multiple blob-formations indicate frequent use of multiple durations of uses.

**Table 1 antibiotics-12-01393-t001:** Characteristics of 3416 residents from Illawarra LTCs between 31 May 2016 to 31 May 2019.

	2016–2017 (*n* = 1592 Residents)	2017–2018 (*n* = 1543 Residents)	2018–2019 (*n* = 1487 Residents)
Number of LTC facilities in cohort	*n* = 18	*n* = 17	*n* = 18
Gender			
Females	63.9 % (*n* = 1018)	62.8 % (*n* = 969)	62.2% (*n* = 925)
Male	28.9% (*n* = 459)	31.0% (*n* = 478)	31.7% (*n* = 472)
Unknown	7.2% (*n* = 115)	6.2% (*n* = 96)	6.1% (*n* = 90)
Age, mean (SD)	85.2 (8.6)	85.1 (8.9)	85.2 (8.9)
Total follow-up time (Per 1000 residents’ days)	585.5	597.5	584.7

**Table 2 antibiotics-12-01393-t002:** Types of antibiotics used in Illawarra LTCs between 31 May 2016 to 31 May 2019.

Antibiotic Types *n* = 10,460	Number of Episodes (%)
Cefalexin	4003 (38.3)
Amoxicillin with clavulanic acid	1190 (11.4)
Amoxicillin	1148 (11.0)
Doxycycline	1024 (9.8)
Trimethoprim	875 (8.4)
Flucloxacillin	570 (5.4)
Trimethoprim with sulfamethoxazole	281 (2.7)
Ciprofloxacin	250 (2.4)
Roxithromycin	250 (2.4)
Clindamycin	232 (2.2)
Metronidazole	109 (1.0)
Nitrofurantoin	104 (1.0)
Cefuroxime	102 (1.0)
Clarithromycin	73 (0.7)
Erythromycin	50 (0.5)
Cefaclor	40 (0.4)
Phenoxymethylpenicillin	38 (0.4)
Norfloxacin	32 (0.3)
Methenamine	29 (0.3)
Dicloxacillin	26 (0.2)

## Data Availability

Restrictions apply to the availability of these data. Data were obtained from Webstercare and are available from the authors with the permission of Webstercare.

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
