# Peer review of "Real-World Data about Commonly Used Antibiotics in Long-Term Care Homes in Australia from 2016 to 2019"

_antibiotics, 2023, doi:10.3390/antibiotics12091393_

Round 1

Reviewer 1 Report

Dear Authors,

In the research, the authors present the consumption (monthly prevalence) of antimicrobial agents in a 3-year observation period among >3000 residents in 17 (18) different LTCH. Although the title suggests antibiotic utilization in LTCH, the authors have mainly focused on antimicrobials recommended for treatment and prophylaxis of UTI, whereas no clinical data were included. Some of the selected antibiotics (amoxicillin and co-amoxiclav) are commonly used for respiratory tract infection as well, also very frequent indication for antimicrobial treatment. Since the authors did not collect the clinical data (indication) for antimicrobial prescription, it is misleading when the authors describe the group as “antibiotics commonly used for UTI”.

The presented manuscript lack discussion about the reasons for high consumption prevalence in observed group. Also, additional data about possible intervention leading to decrease in antimicrobial consumption is missing, which is very important for the future. The study observes the situation but there is no detailed discussion about the observed findings.

Introduction

The authors stated that: “Asymptomatic bacteriuria, the presence of bacteria in urine, and urinary tract infections (UTI) are the most common infections among LTC residents”. It is very important to know (and one of the most common cause of antibiotic misuse) that asymptomatic bacteriuria and the presence of bacteria in urine are NOT infections and do not need antimicrobial treatment (expect in very rare occasions). This could be an important target of the AMS programs in LTCH- to educate medical personal not to treat asymptomatic bacteriuria, which is on the other hand very common in elderly.

Knowing the consumption rate of antimicrobials is the beginning, but for improvement and to explore the real-world praxis it is important to know the indications for antimicrobial use as well. When antibiotic is used correctly (right indication, right dose etc.…) than there is nothing to be done or improve. Therefore, we should not target the reduction in AB consumption by itself, we must rather focus on the misuse of AB and the elements of misuse that can be improved. Therefore, it would be very good to include the clinical data about indication for Ab (for example: right indication/no indication etc.)- this would give important additional value to the collected data.

Results

Since there are many different types of LTCH around the world with residents with different level of comorbidities, it is important to know what type of institutions were included in the study (maybe the Charlson comorbidity score of the residents or other indicators), especially when comparing the results with other published studies.

In Figure 1. the two curves do not differ considerably (as stated by the authors) – at least visually -they look similar to me- are there any statically significant differences?

Also, I cannot see a sharper decrease in overall use of all antibiotics over the study period as stated by the authors. Is there a significant decrease? What is the highest monthly prevalence and what the lowers during the study period?

Inclusion of prophylactic drug methenamine in the study does not make sense, since it is used as prophylaxis only that is used much longer than UTI and consequently the consumption rate and antimicrobial duration is not representable (Ab duration longer than 100 days- probably only prophylaxis).

Figure 2 shows seasonal variation in antimicrobial consumption and as expected there are variations in consumptions of antibiotics used for respiratory tract infection. With more than 10.000 antibiotic episodes in observed period there is just 1 patient receiving single episode if fosfomycin (?)- I would exclude it from Figure 2, since it is not representative.

A lot of data were collected and It would be better to present the consumption of all antibiotics in the observed period in LTCH and not separating a group of “UTI antibiotics”. Comparing this “UTI” group to all antibiotics is therefore not meaningful since consumption of different antibiotic is not in correlation.  

The total number of antibiotic days per 100 resident days has shortened for 2 days during observed period, which I consider as a good sign. Could you comment that?

 Total duration of antimicrobial therapy is nicely presented and it is an interesting finding that the median duration of use was 8 days for most antibiotics. Again, methenamine is not a treatment drug, it is indicated for prophylaxis only and it does not make any sense to compare the duration of prophylaxis with therapy.

Discussion

The authors found considerable high prevalence of antibiotic use among LTC residents in Australia. I would encourage the authors to discuss this higher observed prevalence and the causes for it.

“We found that most antibiotic episodes for the antibiotics commonly used to manage UTI among LTC residents were of a short duration” – I do not agree with this statement, since median duration was 8 days and in Australian guidelines the recommended duration is 3-7 days. It would be very interesting to see how many patients received 3 days of treatment (female cystitis- majority of residents are female), how many 7 or 10 days. This could help in improvement in AB consumption in the future.

I would suggest the authors to improve the study with additional clinical data about indication for antimicrobial use, to discuss the consumption rate in general (not UTI antibiotic consumption separately). I would recommend to add the data about the type of LTCH and structure of the observed residents (comorbidities etc…).

In general, it is important to measure antimicrobial consumption in LTCH residents, since this is a very fragile group, although for implementation of any AMS measures a detailed analysis of consumption would be needed. This study is a very good start, but I would recommend collecting additional data to find the causes for high consumption rates.

Best regards 

Author Response

We would like to thank you for your thorough review of our manuscript and your very useful comments and suggestions. It improved the manuscript and are grateful for this. We have revised the manuscript accordingly. Please see the attachment for a point-by-point reply. The answers in this document refer to the ‘tracked changes’ version of the manuscript. The final manuscript has been agreed on by all authors.

Reviewer 2 Report

I found the study very interesting and I think that the topic is very important. I have some recommendations to authors:

1. Aim of the study could be explained in detail. The study examined not only the trends as more data was extracted and distributed.

2. In the sentence "Our results showed seasonal trends in use of all antibiotics, with antibiotics less frequently initiated in summer (December, January and February) and increasing use in winter (May, June, August)" - months in the brackets are not relevant to the text.

3. Row 77 - 78 presented Table 2, but there is no table 1 before that.

4. The authors have to include the formula used for calculation of utilization per 1000 residents. In section "Results" it is not clear how the results are defined. This should be explained in the methodology section.

5. My personal opinion is that results presented in section "2.4 Duration of use" should be presented in table. It would be more understandable for the readers and final data could be easier compared.

6. It is not absolutely clear how the data was extracted and designed - was antibiotic data from prescriptions taken or from the pharmacy software? Maybe the reference to national (local) software could be provided or additional explanation considering the data provider.

7. I recommend using formulas instead of all the explanations included in the methodology section. Duration, follow up and utilization calculations will be preferable and the design will be more understandable (290- 297; 301-306; 308-315)

Author Response

(The authors gave the same response as above.)

Reviewer 3 Report

Title: Real-world evidence on antibiotic utilization in Long Term 2 Care homes from 2016 to 2019; should include urinary tract infections.

Limitations should be corrected and extended:

Corrected:  The indication or infection site; all are UTI.

Extended: No sensitivities, no blood culture results, no fecal carrier investigation, and no data on evolution, cure, or failure.

In discussion, before strengths and limitations include and discuss:

Bacteremic UTI:

Branwell A. J Clin Microbiol 2023; 61, No 7.

Gomez Belda AB. Geriatr Gerontol Int 2019; 19:1112.

Resistant UTI patogens:

Eure TR. Infect Control Hosp Epidemiol 2021; 42:31.

Diagnosis:

Kuil SD. Clin Infect Dis 2021; 73:e3867.

Latour K. BMC Geriatr 2022; 22:187.

Treatment practices:

Kabbani S. Infect Control Hosp Epidemiol 2022; 43:238.

Antimicrobial Stewardship Australia:

Thursky KA. JAC Antimicrob Resist 2021; 3:dlab166.

Dowson L. Am J Infect Control 2020; 48:261.

Prevention:

Mody L. JAMA Intern Med 2017; 177:1154. 

Author Response

(The authors gave the same response as above.)
